# Behaviour and Anxiety Management of Paediatric Dental Patients through Virtual Reality: A Randomised Clinical Trial

**DOI:** 10.3390/jcm10143019

**Published:** 2021-07-07

**Authors:** Cristina Gómez-Polo, Ana-Aida Vilches, David Ribas, Antonio Castaño-Séiquer, Javier Montero

**Affiliations:** 1Department of Surgery, Faculty of Medicine, University of Salamanca, 37007 Salamanca, Spain; javimont@usal.es; 2Private Dental Practice in Seville, 41003 Seville, Spain; aidavilchesfernandez@gmail.com; 3Department of Stomatology, Faculty of Dentistry, University of Seville, Calle Avicena S/N, 41009 Seville, Spain; dribas@us.es (D.R.); acastano@us.es (A.C.-S.)

**Keywords:** anxiety, behaviour, virtual reality, dental treatment

## Abstract

Clinicians should appreciate the effectiveness of virtual reality (VR) headsets for managing both the anxiety and the behaviour of non-cooperative paediatric patients who require treatment over several dental appointments. The aim of this study was to assess the effectiveness of using a VR headset as a distraction for managing the anxiety and behaviour of paediatric patients during their dental treatment. Eighty patients, aged between five and ten years old and who required dental treatment over three or more appointments, were randomly allocated into two groups. One group used a VR headset during all their appointments, and the other one did not use any distraction technique. The patients were asked to take a Facial Image Scale Test during their first and last appointments to assess their level of anxiety. Additionally, the dentist completed the Frankl Test to quantify the child’s behaviour at the beginning and the end of their treatment. The results obtained, both from the group using the VR headset and from the control group, were compared using the chi-square test. The use of a VR headset during dental treatment significantly reduced anxiety (95% of the children were happy) and improved behaviour (100% positive behaviour) as compared with the control group (40% and 57.5%, respectively). A VR headset can effectively distract a paediatric patient, helping to reduce anxiety and manage behaviour during dental treatment

## 1. Introduction

Anxiety is a normal emotion, basic for our survival and functioning. It helps us to avoid potentially dangerous situations and to prepare ourselves to face challenges. There are multiple definitions of anxiety, one of which describes anxiety as a transitory emotional state of the human organism, characterised by subjective feelings of tension and hyperactivity of the autonomic nervous system. This type of emotional response is externalised in the face of an imminent threat of danger (objective or subjective) and is therefore often presented as a defence mechanism, causing physiological responses related to states of alertness (headache, muscle tension, feelings of suffocation, tachycardia, sweating, and dizziness). Additional signs could also be: diarrhoea, constant movement, nervous tics, excessive sweating, and/or behavioural inhibition. Ordinary stressful life events, such as facing an exam or going to the dentist, can trigger expected forms of anxiety that help prepare humans to overcome challenges [1].

Excessive anxiety can lead to the paediatric patient becoming uncooperative towards their dental treatment, making the process difficult, or even impossible [2,3]. This lack of cooperation due to excessive anxiety leads to negative behaviour, which is the most frequent problem faced by paediatric dentists [4]. It should also be noted that individuals presenting high anxiety in the dental surgery face more time-consuming treatments and increased costs [5,6]. Dental anxiety is also considered to be the most predictive factor of how a child will behave during treatment [7]. Additionally, dental pain is perceived as a multidimensional process involving sensory, cognitive, and emotional components [4]. Several authors have found a strong correlation between dental anxiety and the child’s perception of pain [8]. Dental fear and anxiety affect approximately 15–20% of children [9,10,11].

In these patients who avoid going to the dentist because of anxiety, a vicious circle is created, in which the avoidance of dental treatment leads to dental deterioration and emotions of guilt and inferiority in the patient. This social conflict results in further avoidance, which will lead to the detriment of the individual’s oral health, as well as aesthetic and functional dissatisfaction, determining the person’s lifestyle and compromising their biopsychosocial well-being [12,13]. The theory that dental fear is acquired [14,15,16,17,18,19,20] is reinforced by one’s own negative experiences, unfavourable judgements, and opinions towards the figure of the dentist expressed in the family and/or the immediate environment, which is why any negative comments should be avoided in the presence of the child. Unpleasant dental experiences are not the only element that influences the child’s anxiety and behaviour; it is also necessary to consider their multifactorial origins where genetic (non-modifiable) and environmental (modifiable) factors interact [21,22,23,24]: Identifying the environmental variables that influence children’s behaviour in the dental surgery is essential to be able to control or modify them, and thus to provide a favourable environment for the execution of the dental intervention. Specific training is necessary to treat children, as is the skill of the practitioner, the working systems including protocols adapted to the age of the child, a pleasant atmosphere in the dental surgery, appointment control (ideally in the morning and without the children being kept waiting), or the presence or absence of parents in the dental surgery. There are other factors that cannot be controlled by the dentist: the child’s personality [25] and characteristics, the family’s influence [26], previous negative experiences of the child or of those in the patient’s environment (parents, siblings, or friends) [14,15,16,17,18,19,20], frequency and number of visits to the dentist, cognitive and emotional development, and predisposition to treatment [27].

Several techniques have been described to reduce excessive anxiety [28,29]: (a) Communication techniques: tell-show-do, direct observation, ask-tell-ask, voice control, non-verbal communication, positive reinforcement, and distraction; (b) other basic techniques are, such as parental presence/absence, memory restructuring, and nitrous oxide inhalation; and (c) advanced techniques comprising protective stabilisation, sedation, and general anaesthesia [30,31]. If anxiety can be reduced and the child has a good dental experience, this helps improve adherence to treatment [32,33], reduce the risk and presence of caries [20,33,34], and promote a trusting relationship between patient and dentist.

There are different ways in which behaviour can be modified in relation to a painful experience [35,36]. VR has recently been introduced in the dental field as a technique that could, through distraction during dental treatment, reduce dental pain and anxiety. Distraction is a well-known technique, based mainly on redirecting attention away from the painful stimulus. The “gate control” theory, published by Melzack et al. in 1965 [37], was one of the first reports supporting this concept. The work describes a “gate” located in the medullary dorsal horn, through which painful stimuli pass, which is affected by the activation of A-beta fibres. These fibres are thick and myelinated and inhibit the transmission (close the gate) and conduction of the A-delta and C fibres (permit the painful stimulus to open the gate). Previously published papers [37,38,39,40,41,42] have exploited this theory for reducing the transduction of nociceptive stimuli during dental treatment using a VR headset. At present, the most common distraction techniques used include hypnosis [43,44], music [45,46], audio-visual media [47], and VR [4,9,37,38,39,40,41,42,48,49,50,51], almost all of which are based on distraction, relaxation, imitation, and systematic desensitisation.

These VR devices limit the input of stimuli from the real environment and enhance the input from the virtual environment, decreasing, by perceptual mechanisms, the sensation of presence in the real world and increasing the presence in the virtual environment. Virtual reality glasses and incorporated auditory helmets are the most commonly used components; with them, the subject’s visual and auditory field is practically covered by the virtual information, preventing sensory input from the real dental world (sound of turbines, sight of instruments, needles, injections, etc.) in which the patient is truly immersed. The aim is for the patient to be immersed in and transported to “a parallel reality” that is more pleasant and for them, one in which they are not capable of perceiving unpleasant dental elements. The simultaneous stimulation of sight and hearing or of hearing, sight, and touch can enhance the experience, producing a more effective distraction [52]. Currently, VR headsets are more affordable and can create a higher level of immersion in a particular situation than traditional audio-visual media. They allow the user to interact with stimuli and to escape from the real world, creating a sensorial protection barrier.

Despite the number of studies on the subject, there is no universally accepted method for controlling anxiety and managing the behaviour of paediatric patients, especially when treating non-cooperative patients who neglect their oral health care [53].

Several papers have studied the use of a VR headset to distract children during dental treatment [38,48,49,50,51,54], even though the results have shown this technique significantly reduces the symptoms of dental anxiety and misbehaviour, except in the case of the study by Sullivan et al. [54]. Furthermore, until now, the influence of this technique on factors related to oral health, such as the frequency of tooth brushing or oral health, has not been studied. The VR headset is expected to not only reduce anxiety and improve patient behaviour but also to reduce treatment time, making the dentist’s job easier. This study aimed to assess the effectiveness of using a VR headset as a distraction technique to reduce anxiety and improve the behaviour of children during dental treatment. In addition, the main socio-demographic, clinical- and parent-related factors were analysed.

## 2. Materials and Methods

### 2.1. Participants

The final sample size was determined using the data distribution and means of the dental anxiety score of the first 12 participants to determine that 38 participants were needed to obtain a power of 0.80% and an alpha error of 0.05% for detecting a true difference in means between the test and the reference group of 0.6. To compensate for sample attrition, a clinical study involving two sets of patients was performed: a control group (*n* = 40) and an experimental VR group (*n* = 40). The VR group consisted of randomly selected, healthy patients between 5 and 10 years of age, who lived in the province of Seville (Spain) and needed a minimum of three appointments to undergo conservative dental treatment (fillings). To effectively randomize the intervention, we used sequentially numbered, sealed opaque envelopes, with aleatory numbers that were selected by the participant at the first appointment and latter allocated to test (pair numbers) or control (odd numbers) groups accordingly. All dental operations were performed under topical anaesthesia and subsequently under intravenous anaesthesia. No pharmacological measures were used for the management of behaviour during the dental treatment. In addition, these patients underwent a distraction technique during treatment using a VR headset. The design of this study was previously approved by the Research Ethics Committee the Virgen Macarena and Virgen del Rocio University Hospitals (C.P. AVF–C.I. 0949-N-17). Informed consent was obtained from all of the participants’ parents or legal guardians.

### 2.2. Outcomes

The data collected for each patient included socio-demographic information (age, gender, and address) and clinical information (medico-dental history, parafunctional habits, frequency of teeth brushing, and oral health status in both temporary and permanent dentition). The patients’ oral health status was summarised by recording the total number of decayed and filled temporary teeth (DFT index) and the total number of decayed, missing, and filled permanent teeth (DMFT index).

The data collected through the use of questionnaires were: (a) The anxiety level of the child’s mother or father, registered during the first appointment using the Corah Dental Anxiety Scale (CDAS) translated into Spanish by Pal-Hegedüs C, et al., which is a questionnaire consisting of four questions, each with five possible answers (scores ranging from 1 to 5), and a Likert scale, which ranges from 4 points (not anxious) to 20 points (extremely anxious). This scale is mostly used to quantify the degree of dental anxiety [55]. (b) Each paediatric patient took the Facial Image Scale Test [56], which is a visual test used to rate anxiety, at the end of the first and last appointments. This test is intended for children to identify their own level of anxiety during invasive treatments by means of “faces” with expressions ranging from very happy to very sad, used in patients from 3 to 18 years of age. The FIS scale ratings included being 1—very happy, 2—happy, 3—neutral, 4—sad, and 5—very sad. It is claimed that the FIS is a valid and reliable measure of dental anxiety in young children (Figure 1). (c) The dentist completed the Frankl Test [57] to quantify the child’s behaviour at the end of the first and last appointments, as this is the most widely used questionnaire and is considered reliable. The Frankl test [57] was rated as 1—definitely positive (smiles/laughs, cooperates, enjoys themselves, and even shows interest in the treatment), 2—positive (accepts the treatment and obeys but is cautious and anxious), 3—negative (accepts the treatment with difficulty, does not engage and is distant, absent), and 4—sad (completely refuses treatment, yells, cries, and does not cooperate).

The VR group was distracted using the Zeiss Cinemizer (Carl Zeiss AG, Oberkochen, Germany) VR headset during dental treatment, which is an individual headset that has earphones and a mask incorporated (Figure 2). The cartoons or children’s movies shown to the children were age appropriate and selected together with each child’s parents. The cartoons included Peppa Pig, SpongeBob SquarePants, Paw Patrol, Dora the Explorer, and Doc McStuffins, and the films included Frozen, Despicable Me, Finding Dory, and Moana. The patients who were randomly allocated to the control group were treated by the same dentist and the same auxiliary staff in the same dental surgery and were evaluated using the same tests, but no distraction technique was used. Both the dentist and the auxiliary staff who attended all participants were trained in paediatric dentistry techniques, and the basic care protocol used was the “tell-show-do” technique in a context of positive motivation and optimism.

### 2.3. Statistical Analysis

The descriptive statistical analysis of the variables included mean, standard deviation for the continuous variables, and relative and absolute frequencies for the categorical variables. Since the variables used (anxiety and behaviour) were based on a Likert scale and not pure quantitative variables, the Wilcoxon signed-rank nonparametric test was applied to assess the effect of time (between the first and last appointments) within the group. The chi-squared test was also used to measure the association between two categorical variables. Finally, the value for statistically significant differences was established as a *p* value of <0.05. The predictability of children’s final dental anxiety and final behaviour was studied using a linear forward stepwise regression model (forward stepwise regression analysis). The data analysis software used was SPSS (Statistical Package for Social Sciences) version 19 (Chicago, IL, USA).

## 3. Results

In this study, both the control and VR groups were comparable in terms of their socio-demographic and clinical data (Table 1). The sample was composed of more girls than boys (56.3% compared with 43.8%); 95% of the children were aged between 6 and 10 years old, lived in a rural environment (58.8%), and brushed their teeth daily (91.2%). Most of the children did not have any habits harmful to their oral health (77.5%), but there was an elevated prevalence of caries in both temporary (90%) and permanent teeth (65%). The patients received conservative dental treatment, on average, during a total of 5.0 ± 1.9 appointments. No significant differences were found between control and VR groups with respect to any of the variables mentioned above.

According to the Corah tests (CDAS), most of the parents were relaxed (46.3%) or slightly worried (28.8%) about their child’s dental treatment (Table 2), although there were no differences found between the control and the VR groups. Additionally, the level of anxiety at the time of the first appointment was comparable between groups, and only 10% of children were sad or very sad (Table 2). However, child behaviour at baseline (in the first appointment) was significantly worse for the children in the VR group (25% behaved negatively) than the control group (10% behaved negatively), as shown in Table 2.

The effect of the VR headset on the patient’s level of anxiety (FIS test) and behaviour (Frankl test) can be observed by comparing the data recorded during the last dental appointment (Table 2). Children treated together with the use of VR were significantly happier and better behaved than those in the control group. It was observed that both groups had a similar state of mind during the first appointment, at which time 60% of the patients considered themselves to be happy or very happy. However, the children using the VR headset experienced less anxiety, with 95% of children describing themselves as being happy or very happy during their last appointment (Table 2). A statistically significant decrease in child anxiety was observed in the group that used VR as distraction, as well as an improvement in final behaviour. Intra-group comparisons with the Wilcoxon signed-rank tests show that while behaviour and anxiety improved significantly in the test group between the first and the last appointment (*p* < 0.001), both behaviour and anxiety worsened in the control group (*p* < 0.001).

In order to determine which independent variables (age, gender, DMFT index, DFT index, parents’ dental anxiety, number of appointments, group (control vs. VR group) baseline anxiety, and baseline behaviour) are able to significantly predict the dependent variables Final Child Anxiety (FIS test) and Final Child Behaviour (Frankl Test), a statistical analysis called stepwise forward regression analysis was conducted. Only those variables found to be significant (*p* < 0.05) for final child anxiety (Table 3) and final child behaviour (Table 4) were included as independent predictor variables in the linear regression model.

With respect to final child anxiety, we observe that the proposed model, with only two independent variables in Table 3, was able to explain 65% (*R*^2^ = 0.65) of all the variability of the response data on anxiety. The predictor variables were group membership (whether or not VR is used) and initial anxiety state. Wearing a VR headset decreased child anxiety values between 1.2 and 1.7, while initial anxiety was proportional to final anxiety and increased with each category by between 0.35 and 0.64.

The independent predictor variables that were found to be significant (*p* < 0.05) for predicting the child’s final behaviour were group membership (VR vs. control), initial behaviour, and number of appointments (Table 4). The model for predicting the child’s final dental behaviour had an explanatory power of 63%. Similar to what was observed for anxiety, wearing a VR headset improved child behaviour values between 1.2 and 1.7, while initial behaviour conditions final behaviour, changing with each category between 0.35 and 0.64 of final behaviour.

## 4. Discussion

Methods to decrease anxiety and better manage behaviour within the dental surgery focus on avoiding unpleasant and unproductive behaviours, creating a pleasant and trusting environment that can facilitate the performance of treatment, and developing positive attitudes towards future dental care. In general, the participants presented a moderately good level of oral hygiene, with 86.3% brushing their teeth once or twice a day. In both VR group and control groups, it was found that children who brush their teeth less often also suffer from higher levels of anxiety and misbehave more often.

In most studies, the most commonly employed technique to obtain the child’s cooperation is “tell-show-do” [30,58,59,60,61], as used in the present research, followed by the positive reinforcement technique [59,60,62], also used by the trained professional who attended the children. Other authors point to distraction techniques (73%) as the second most used technique [63]. In order to use the “tell-show-do” and positive reinforcement techniques, it is not necessary to have any specific equipment, which is essential for techniques based on distraction with Virtual Reality. This work assessed how child anxiety and behaviour can be modified with the use of a VR headset, as well as the main modulating factors. Some studies using VR systems in dentistry only consider age and sex [38,39], while others consider physiological changes [39,54,64] such as pulse rate (which reduced during the study, according to all the authors). This study did not take pain perception or physiological changes into account. Furthermore, as questions or suggestions concerning pain could lead to disruptive behaviour, they were intentionally avoided, especially considering that the patients were children. A pulse oximeter had been used in a previous pilot study to monitor changes in pulse rate and oxygen saturation; however, the use of the apparatus seemed to frighten the children due to their young age. As a result, physiological changes were not monitored during this study to avoid conditioning the patients.

The prevalence of dental anxiety decreases as children’s ages increase, and the frequency was higher in females than in males [6,65]. In contrast, the present study found neither age nor gender to be independent predictors of dental anxiety or behaviour. The increase in the number of visits to the dentist does seem to have a negative influence on the child’s behaviour, in line with what has been published by Jeddy et al. [65]. However, the authors describe the process and explained that during the first visits there is more anxiety and fear than during the last ones, as they are acquiring acceptance mechanisms and learning to distinguish between procedures that produce tension and those that do not [66]. The questionnaires used to assess dental anxiety and behaviour were chosen because they were easy to answer due to their simplicity. A literature review [30,31,32,35] was initially carried out and revealed a meta-analysis [36] that examined the relationship between parent mental status and a child’s fear of the dentist. The meta-analysis by Themmessl-Huber et al. [36] studied the relationship between parents and the dental fear of younger (0 to 8 years old) and older children (8–10 years old). In the present study, anxiety was measured using the visual Facial Image Scale test [31], the DAS [30] questionnaire, and the Frankl behaviour assessment rating scale [32]. In the older group of children, the 8- to 10-year-olds, we found the results were different when using these particular assessment methods than those obtained in other studies using undetermined methods. The VR Zeiss Cinemizer OLED system is available on the market and can be easily acquired. Furthermore, the participants in this study received dental treatment that is commonly carried out in paediatric dentistry. Based on our results, the patient’s initial anxiety level is a predictor of their final dental anxiety, and their initial behaviour is a significant predictor of their final behaviour, but it is not influenced by parental anxiety status. Using a VR headset significantly reduces anxiety and improves children’s behaviour. Several research studies also concluded that a VR headset is beneficial in optimising children’s collaboration and making the experience more satisfying [6,48,49,50,51,67].

A recent published meta-analysis shows that VR significantly reduced dental anxiety in children because it was an effective method of distraction suitable for a wide variety of dental treatments. Additionally, the heterogeneity of the research design made direct comparisons difficult, as there were wide differences in terms of the age ranges studied, the questionnaires used, and the type of treatment carried out. The test selected to quantify dental anxiety in children was FIS, a very intuitive visual test (Figure 1), but which prevented direct comparison of scores from non-visual tests such as MDAS [49,50,51], VCARS [48], and FLACC [67]. However, all agree on the usefulness of the VR headset, regardless of the brand, manufacturer, or features of the VR device in reducing dental anxiety.

The use of a VR headset has been reported to reduce blood pressure, pulse rate, and pain indices in patients undergoing periodontal scaling and root planing procedures [68,69]. Our findings are not supported by any other series of studies by Bentsen et al. where a VR headset did not alter perceived pain intensity [70] or dental scaling in adult patients [71]. More research is needed to explore whether the effectiveness of this type of distraction may depend on patient-related factors, such as personality attributes, previous experiences, coping styles, or sense of predictability or control. However, a systematic review in children stated that there is evidence that audio-visual distraction is effective in managing anxiety in children, as discussed above [39,72]. Not to be forgotten is the opinion of the children themselves: most patients (74%) stated that they would prefer using the VR headsets during future appointments involving dental fillings [41]. High-quality randomised clinical trials are needed to determine the (relative) effectiveness of these interventions in reducing anxiety and improving child collaboration at the dental surgery.

The ages considered in this study ranged from 5 to 10 years, similar to those considered in the study performed by Ram et al. [39]. Although the sample consisted of children, they were in fact old enough to complete the questionnaire used to assess their own level of anxiety and to take the Facial Image Scale [56] visual test. Additionally, they were able to understand the use of the VR headset and not to feel intimidated or scared by covering their eyes in front of unknown people, without their parents being present and surrounded by unpleasant smells and unfamiliar instruments. Other research related to paediatric patients chose a wider range of ages, such as the study by Hoge et al. [41], which included patients from 4 to 16 years old, and the study by Hoffman et al. [68], whose sample population ranged from 9 to 32 years old.

Cattell et al. [73] also carried out a literature review and came to the conclusion that any highly consistent measurements of anxiety are based on self-assessment. The Facial Image Scale [56], which is a visual test, was chosen for this reason and allows the patient to rate their own level of anxiety by choosing from five different faces. Dental anxiety has been studied by several authors, but the paper by Aminabadi et al. [38] specifically concentrates on children. The sample included 120 four- to six-year-olds with their first two molars decayed. In addition, these children had not suffered from any type of anxiety disorder during their first dental appointment, according to the Screen for Child Anxiety Related Disorders Questionnaire [74]. This sample was divided in two groups: one group was treated during the first appointment using VR but not during the second appointment, while in the other group the use of VR was the opposite. Statistically significant differences were found between the two appointments for both groups. The first group showed mean values of anxiety during the first appointment (with VR) of 12.58 ± 1.01 and 17.68 ± 1.25 during the second appointment (without VR). These values represent a statistically significant difference with respect to the increase in anxiety. The anxiety values of the second group were 18.25 ± 1.02 during the first treatment session (without VR), but these values dropped to 13.20 ± 1.00 during the second (with VR distraction). A statistically significant difference between the two appointments was found for both groups (*p* < 0.001). This decrease in anxiety is slightly greater than the one found in the study by Aminabadi et al. [38]. However, it should be mentioned that the methods employed to assess anxiety were different.

There are discrepancies as to whether the DMFT and DFT indices are related to the presence of dental anxiety [75]; however, other studies, like the present results, have found no relationship between dental anxiety and a higher DMFT index [76]. Some components inherent to dental treatment cause more dental anxiety than others, with most authors naming the sight of a needle and its sensation and the sight of the turbine, its sensation, and the noise it produces as the main sources [77,78,79]. Another point of view is presented by Weinstein et al. [80], who point out that procedures that only apply the principle of distraction do not seem to be useful in paediatric dentistry and that a combination of positive reinforcement, negative reinforcement, voice control, and cognitive-behavioural techniques should be applied [81,82].

Other types of treatments to reduce anxiety and improve behaviour could be child hypnosis, although there is still no clear evidence [44]. Hugly et al. [83] were among the first authors to consider music as a stress reliever. Additionally, it increases relaxation through muscular hypotonia [84]. Other research, on the contrary, stated that distraction by music is not an effective method to reduce anxiety and pain or to improve behaviours [45]. Peretz et al. [85] describe the use of magic tricks as a possible alternative to relax the environment and encourage child collaboration.

The second objective of our study was to assess behaviour. Some authors, such as Hoge et al. [41] and Ram et al. [39], have found less disruptive behaviour in their experimental groups, as was the case in this study. By contrast, the study performed by Sullivan et al. [54] on a sample size of 26 patients did not find statistically significant differences between the experimental and control groups in terms of anxiety and managing behaviour using VR headsets during dental treatment.

Creating positive memories is a very important aspect of paediatric restorative dentistry. Tanja-Dijkstra et al. [40] suggest that distraction through VR is effective not only during the experience but also after the dental treatment has finished. The use of VR creates a profound illusion that allows the patient to enter into a virtual world and, at the same time, stimulates their senses. Dahlquist et al. [86] obtained similar results to our study using VR, where it was found that VR seemed more effective with older rather than younger patients when compared with other simple distraction techniques. The positive experience of a visit to the dentist can be very beneficial, not only in terms of oral health, but it also reinforces the child’s confidence and reaffirms his or her ability to face complex or challenging situations and come out victorious, which will probably lead to the next dental visit being perceived with less anxiety.

## 5. Conclusions

The use of a VR headset during dental treatment significantly reduces anxiety, with 95% of the patients feeling very happy during their last appointment, and it also significantly improves behaviour, with 100% of the patient being rated as behaving positively or very positively. Baseline anxiety and behavioural situations condition final anxiety and behaviour independent of the effect of distraction.

## Figures and Tables

**Figure 1 jcm-10-03019-f001:**
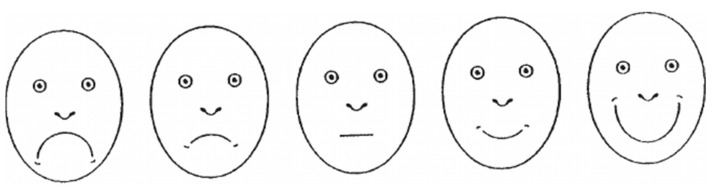
The Facial Image Scale (FIS) [57] used in this study.

**Figure 2 jcm-10-03019-f002:**
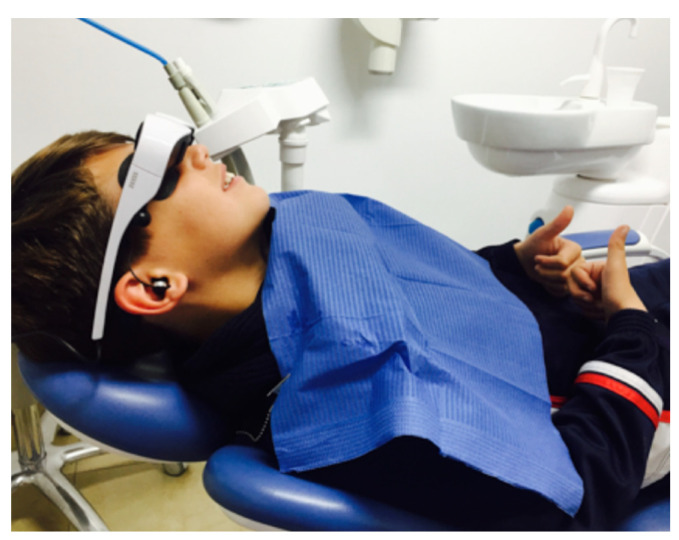
The VR headset used by the VR group during dental treatment.

**Table 1 jcm-10-03019-t001:** Description of the study sample (*n* = 80) and both VR group and control subgroups (*n* = 40) regarding sociodemographic, behavioural, and clinical variables.

Variables	Total (*n* = 80)	Control (*n* = 40)	VR Group (*n* = 40)	*p* Value
Sociodemographic				
**Age (years)**	***n* (%)**	***n* (%)**	***n* (%)**	
5 years	4 (5.0)	1 (2.5)	3 (7.5)	0.85 (Chi^2^)
6 years	16 (20.0)	10 (25.0)	6 (15.0)	
7 years	17 (21.3)	10 (25.0)	7 (17.5)	
8 years	9 (11.3)	3 (7.5)	6 (15.0)	
9 years	15 (18.8)	5 (12.5)	10 (25.0)	
10 years	19 (23.8)	11 (27.5)	8 (20.0)	
	**Mean (sd)**	**Mean (sd)**	**Mean (sd)**	
**Mean Age (years)**	7.9 (1.6)	7.9 (1.7)	8.0 (1.6)	0.78 (*t*-Student)
**Gender**	***n* (%)**	***n* (%)**	***n* (%)**	
female	45 (56.3)	19 (47.5)	26 (65.0)	0.115 (Chi^2^)
male	35(43.8)	21 (52.5)	14 (35.0)	
**Residence**	***n* (%)**	***n* (%)**	***n* (%)**	
Urban	33 (41.3)	18 (45.0)	15 (37.5)	0.496 (Chi^2^)
Rural	47 (58.8)	22 (55.0)	25 (62.5)	
**Behavioural Variables**				
**Frequency of brushing teeth**	***n* (%)**	***n* (%)**	***n* (%)**	
Less than once/day	7 (8.8)	4 (10.0)	3 (7.5)	0.982 (Chi^2^)
once/day	33 (41.3)	16(40.0)	17(42.5)	
twice/day	36 (45.0)	18(45.0)	18 (45.0)	
3 times/day	4 (5.0)	2(5.0)	2(5.0)	
	**Mean (sd)**	**Mean (sd)**	**Mean (sd)**	
Number of daily brushings	1.5 (0.7)	1.4 (0.7)	1.5 (0.7)	0.68 (*t*-Student)
**Harmful habits**	***n* (%)**	***n* (%)**	***n* (%)**	
None	62 (77.5)	32 (80.0)	30 (75.0)	0.592 (Chi^2^)
Nail biting	13 (16.3)	5 (12.5)	8 (20.0)	
Thumb/dummy sucking	2 (2.6)	2(5.0)	0 (0.0)	
Baby bottle	2.0 (2.5)	1.0 (2.5)	1.0 (2.5)	
Oral breathing	1.0 (1.3)	0.0 (0.0)	1.0 (2.5)	
**Clinical**				
**Caries prevalence**	***n* (%)**	***n* (%)**	***n* (%)**	
Permanent dentition	52 (65.0)	25 (62.5)	27 (67.5)	0.639 (Chi^2^)
Temporary dentition	72 (90.0)	36 (90.0)	36 (90.0)	1.0 (Fisher)
	**Mean (sd)**	**Mean (sd)**	**Mean (sd)**	
DMFT index	1.9 (1.8)	1.7 (1.6)	2.2 (1.9)	0.71 (*t*-Student)
DFT index	4.4 (3.0)	3.9(3.1)	5.0 (2.9)	0.56 (*t*-Student)
Number of appointments	5.0 (1.9)	5.1 (1.8)	5.0 (2.0)	0.861 (*t*-Student)
**Baseline Parent Anxiety (CDAS)**	***n* (%)**	***n* (%)**	***n* (%)**	
Relaxed	37 (46.3)	18 (45.0)	19 (47.5)	0.20 (Chi^2^ = 6.06)
Worried	23(28.8)	14 (35.0)	9 (22.5)	
Moderate Anxiety	8 (10.0)	1 (2.5)	7 (17.5)	
Severe Anxiety	9 (11.3)	5 (12.0)	4 (10.0)	
Dental Phobia	3 (3.8)	2 (5.0)	1 (2.5)	

**Table 2 jcm-10-03019-t002:** Comparison of the children’s dental anxiety (FIS Test) and the behaviour of children (Frankl Test) at the end of the first appointment (baseline) and at the end of the last appointment (final) between VR (*n* = 40) and control subgroups (*n* = 40).

Child Anxiety (FIS Test) [58] *n* (%)					Child Behaviour (Frankl Test) [59] *n* (%)				
	Control (*n* = 40)		VR Group (*n* = 40)			Control (*n* = 40)		VR Group (*n* = 40)	
	BASELINE ^a^	FINAL *^,b^	BASELINE ^a^	FINAL *^,b^		BASELINE *^,c^	FINAL *^,d^	BASELINE *^,c^	FINAL *^,d^
Very Happy	9 (22.5)	4 (10)	8 (20.0)	31 (77.5)	Definitely positive	19 (47.5)	2 (5.0)	9 (22.5)	34 (85.0)
Happy	16 (40.0)	12 (30)	16 (40.0)	7 (17.5)	Positive	17 (42.5)	21 (52.5)	21 (52.5)	6 (15.0)
Neutral	11 (27.5)	15 (37.5)	12 (30.0)	2 (5.0)	Negative	2 (5.0)	12 (30)	9 (22.5)	0 (0)
Sad	3 (7.5)	8 (20.0)	3 (7.5)	0 (0)	Definitely negative	2 (5.0)	5 (12.5)	1 (2.5)	0 (0)
Very Sad	1 (2.5)	1 (2.5)	1 (2.5)	0 (0)					

^a^: *p* value = 0.20 (Chi^2^ = 0.1); ^b^: *p* < 0.001 (Chi^2^ = 41.1); ^c^: *p* value < 0.05 (Chi^2^ = 8.78); ^d^: *p* < 0.001 (Chi^2^ = 53.8); *: Significant within group pre-post differences after Wilcoxon signed-rank tests (*p* < 0.001).

**Table 3 jcm-10-03019-t003:** Linear regression analysis for forecasting the final child anxiety (FIS test) outcomes as a function of the group (control vs. VR group) and baseline anxiety (*n* = 80).

Dependent Final Child Anxiety	Standardised β	Error	T	*p* Value	Lower CI 95%	Upper CI 95%
Predictors						
Group (control vs. VR group)	−0.7	0.1	−10.5	<0.001	−1.7	−1.2
Baseline anxiety	0.4	0.1	6.7	<0.001	0.35	0.64

*F* = 77.0; *p* < 0.001; Corrected *R*^2^ = 0.65.

**Table 4 jcm-10-03019-t004:** Linear Regression Analysis for forecasting the final child behaviour (Frankl test) outcomes as a function of the group (control vs. VR group) and baseline behaviour and number of appointments (*n* = 80).

Dependent Final Child Behaviour	Standardised β	Error	T	*p* Value	Lower CI 95%	Upper CI 95%
Predictors						
Group (control vs. VR group)	−0.7	0.12	−11.4	<0.001	−1.7	−1.2
Baseline Behaviour	0.8	0.08	3.7	<0.001	0.14	0.46
Number of appointments	0.2	0.04	2.1	<0.05	0.006	0.13

F = 45.8; *p* < 0.001. Corrected R^2^ = 0.63.

## Data Availability

Not applicable.

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
