# Peer review of "Behaviour and Anxiety Management of Paediatric Dental Patients through Virtual Reality: A Randomised Clinical Trial"

_jcm, 2021, doi:10.3390/jcm10143019_

Round 1
Reviewer 1 Report
Dear Authors,
Dentophobia is an important issue in treating patients, especially children. The aim of the study was to assess the effectiveness of using a VR headset as a distraction for managing the anxiety and behaviour of paediatric patients during their dental treatment.
Punctuation and editorial errors in the text should be corrected. Highlights need to be removed from materials and methods, results and discussion. p value should be written in italics.
Different notation of p value: p<.008; p=.05; p=0.008
Rounding to the same digit after the point.
Table 1 - should be done in accordance with JCM guidelines. If n (%) is written, we avoid % in individual lines. Need to standardize.
4 (5.0%) vs. 4 (5.0)
Table 2, 3 and 4 - should be done in accordance with JCM guidelines.
English language and style are fine.
Cianetti S, Lombardo G, Lupatelli E, Pagano S, Abraha I, Montedori A, Caruso S, Gatto R, De Giorgio S,
Salvato R. Dental fear/anxiety among children and adolescents. A systematic review. Eur J Paediatr Dent. 2017
Jun;18(2):121-130. doi: 10.23804/ejpd.2017.18.02.07. PMID: 28598183.
vs.
Zhang, J.; Yu, X.; Guo, P.; Firrman, J.; Pouchnik, D.; Diao, Y.; Samulski, R.J.; Xiao, W. Satellite Subgenomic Particles Are Key Regulators of Adeno-Associated Virus Life Cycle. Viruses 2021, 13, 1185.
To sup up, article can be accepted afer corrections.
Author Response
Please, see attached file.

Reviewer 2 Report
Interesting and well written article
Author Response
Thank you for your positive comments.
Regards
This manuscript is a resubmission of an earlier submission. The following is a list of the peer review reports and author responses from that submission.
Round 1
Reviewer 1 Report
Dear Authors,
Thank you for the opportunity to review the paper entitled “Behavior and anxiety management of pediatric dental patients through virtual reality: a randomized clinical trial”. The study was aimed to assess the effectiveness of using a VR headset as a distraction for managing anxiety and behavior during the dental treatment of pediatric patients. The paper presents a very interesting field of study, but the scientific quality of the manuscript is low. Editing errors should be noted as well as aspects related to data analysis and presentation of results. I believe that in its current form the article is not suitable for publication. Below is a list of my comments
Major:
- How randomization and allocation to groups were performed is not indicated.
- Sample size calculations are not indicated
- The tables presented are very messy. Please improve them to readable ones
- In the results presented (tables, results section) I do not see the results of repeated measures ANOVA. In Tables 2 and 3, only the results between groups at consecutive time points, before treatment (Table 2), and at the last treatment appointment (Table 3) are summarized. How did anxiety and behavioral values change for children within the group in subsequent meetings? It was described, after all, that there were at least 3 appointments with children.
- No discussion of the results obtained is presented in the paper. It was decided to focus only on the results of other authors - a big oversight. Why did VR act as a distractor?
Minor:
- I propose to change the nomenclature from "virtual headset" to "VR headset", the headset wasn't virtual, isn't it so?
- I'm not entirely convinced that this phrase is correct "improves anxiety" (line 65) - in my opinion, it should sound like a reduction of anxiety symptoms.
- In the paper, several times multiple spaces are introduced
- Please enter abbreviations the first time you use a word without re-translating it: i.e. “virtual reality (VR)” (lines 63, 80)
- Information about clinical trial registration is missing
- For better readability of the paper, I suggest dividing Chapter 2, Materials and Methods, into subsections e.g. 2.1 Participants; 2.2 Outcomes; 2.3 Statistical analysis…
- The discussion is incoherent, it's hard to follow, I suggest dividing it into a logical sequence of topics, rather than "throwing in" more works
Author Response
Thank you for the opportunity to review the paper entitled “Behavior and anxiety management of pediatric dental patients through virtual reality: a randomized clinical trial”. The study was aimed to assess the effectiveness of using a VR headset as a distraction for managing anxiety and behavior during the dental treatment of pediatric patients. The paper presents a very interesting field of study, but the scientific quality of the manuscript is low. Editing errors should be noted as well as aspects related to data analysis and presentation of results. I believe that in its current form the article is not suitable for publication. Below is a list of my comments
Major:
- How randomization and allocation to groups were performed is not indicated.
To effectively randomize the intervention we used sequentially-numbered, sealed, opaque envelopes, containing aleatory numbers; participants selected a number at their first appointment, which was later allocated to the test (even numbers) or control (odd numbers) group accordingly.
- Sample size calculations are not indicated
There was not a sample size calculation a priori; instead we used a greater sample size than existing comparable studies (Sullivan et al. on a sample size of 26 patients, Shetty et al ,n=60, Nunna et al, n= 35, Niharika et al, n= 18, Mitrakul et al, n= 21, Asvanund et al, n=26, Asl-Amonabadi et al, n=60, Al-khotani n=28). This convenience sample size (n=40 per subgroup) has been demonstrated to be efficient enough to find a significant difference between pre and post scores, using both the Frankl and the Phobia Scales.
- The tables presented are very messy. Please improve them to readable ones
All tables have been improved. Linear regression has been added.
- In the results presented (tables, results section) I do not see the results of repeated measures ANOVA. In Tables 2 and 3, only the results between groups at consecutive time points, before treatment (Table 2), and at the last treatment appointment (Table 3) are summarized. How did anxiety and behavioral values change for children within the group in subsequent meetings? It was described, after all, that there were at least 3 appointments with children.
The intra-group comparisons have been carried out with the Wilcoxon signed Rank test for both anxiety and behavioural scales. The results are shown in Table 3 and commented in the text.
Intra-group comparisons using the Wilcoxon Signed Rank Tests show that while behaviour and anxiety improve significantly in the test group between the first and the last appointment (p<0.001), both behaviour and anxiety worsen in the control group (p<0.001).
- No discussion of the results obtained is presented in the paper. It was decided to focus only on the results of other authors - a big oversight. Why did VR act as a distractor?
There are different factors that contribute to increase the sensation of presence in a virtual reality environment. Some of them are perceptual and others are motor. In dentistry, we will focus on the perceptual ones, since the subject must remain seated and still in the dental chair (he/she is limited to seeing what is happening through the VR device and does not interact with movements). These VR devices limit the input of stimuli from the real environment and enhance the input from the virtual environment, decreasing, by perceptual mechanisms, the sensation of presence in the real world and increasing the presence in the virtual environment. Virtual reality glasses and incorporated auditory helmets are the most commonly used components; with them, the subject's visual and auditory field is practically covered by the virtual information, preventing sensory input from the real dental world (sound of turbines, sight of instruments, needles, injections, .... ) in which the patient is truly immersed. The aim is for the patient to be immersed in and transported to "another parallel reality" that is more pleasant and that is not capable of perceiving aversive dental elements.
However, the Discussion section has been expanded and reformulated to make it clearer for the reader
Minor:
- I propose to change the nomenclature from "virtual headset" to "VR headset", the headset wasn't virtual, isn't it so?
OK, nomenclature has been changed to “VR headset”
- I'm not entirely convinced that this phrase is correct "improves anxiety" (line 65) - in my opinion, it should sound like a reduction of anxiety symptoms.
OK, the phrase has been amended
- In the paper, several times multiple spaces are introduced
Spaces have been removed
- Please enter abbreviations the first time you use a word without re-translating it: i.e. “virtual reality (VR)” (lines 63, 80)
The “VR headset” terminology has been standardized.
- Information about clinical trial registration is missing
Institutional Review Board Statement: “The study was conducted according to the guidelines of the Declaration of Helsinki, and approved by the Research Ethics Committee of Virgen Macarena and Virgen del Rocio University Hospitals, Seville, Spain (C.P. AVF – C.I. 0949-N-17/ 2007).
- For better readability of the paper, I suggest dividing Chapter 2, Materials and Methods, into subsections e.g. 2.1 Participants; 2.2 Outcomes; 2.3 Statistical analysis…
OK, done
- The discussion is incoherent, it's hard to follow, I suggest dividing it into a logical sequence of topics, rather than "throwing in" more works
The Discussion section has been expanded and reformulated to make it clearer for the reader
Reviewer 2 Report
The present study aims to analyze the effectiveness of the use of a virtual reality headset as a distraction for the management of anxiety and behavior during the dental treatment of pediatric patients.
Although it is a topic of interest today where audiovisual media are used on a daily basis, authors still need to work on the article for it to be ready for publication. Some suggestions to improve the manuscript could be:
Title
The title is correct and descriptive of the work.
Introduction
The introduction is appropriate according to the topic addressed. It is descriptive of anxiety and behavior during children's dental treatment, as well as the main techniques to overcome them.
Data on the prevalence of treatments that need to be interrupted or postponed due to children's behavior or anxiety should be entered.
The authors note that there are “a hand full of studies have used a virtual reality (VR) headset to distract children during dental treatment” [11-13,16]. However, other references included by the authors also use virtual reality for distraction at the dentist. Likewise, I recommend reading the following literature among others:
- Nunna M, Dasaraju RK, Kamatham R, Mallineni SK, Nuvvula S. Comparative evaluation of virtual reality distraction and counter-stimulation on dental anxiety and pain perception in children. J Dent Anesth Pain Med. 2019;19:277-288.
- Niharika P, Reddy N V, Srujana P, Srikanth K, Daneswari V, Geetha K S. Effects of distraction using virtual reality technology on pain perception and anxiety levels in children during pulp therapy of primary molars. J Indian Soc Pedod Prev Dent 2018;36:364-9
- Vabitha Shetty, Lekshmi R Suresh, Amitha M Hegde; Effect of Virtual Reality Distraction on Pain and Anxiety During Dental Treatment in 5 to 8 Year Old Children. J Clin Pediatr Dent 1 2019; 43: 97–102
- Al-Khotani A, Bello LA, Christidis N. Effects of audiovisual distraction on children's behaviour during dental treatment: a randomized controlled clinical trial. Acta Odontol Scand. 2016;74(6):494-501
Material and Methods
The methodology used in the clinical trial must be detailed in greater depth, especially as regards dental treatment. The need for preventive, restorative, pulp treatment, extraction or any other treatment is not specified, even knowing the existence of a wide variability between the different treatments for children's anxiety or behavior. Also, an assessment of pain in the different procedures would be desirable.
On the other hand, an assessment of the patient's previous experiences in dental treatment would be interesting.
Results
The results of the study, together with the statistics analyzed, are presented acceptably.
The results of the DFT and DMFT index are not presented, keys for evaluating the existence of previous treatments.
Line 148 must be deleted.
Discussion
The discussion shows an interesting analysis of the results of the study in comparison with other studies present in the literature.
A comment with other common techniques to reduce anxiety and improve behavior such as counter-stimulation would be necessary.
Finally, tables should be modified and references rewritten according to journal guidelines.
Author Response
The present study aims to analyze the effectiveness of the use of a virtual reality headset as a distraction for the management of anxiety and behavior during the dental treatment of pediatric patients. Although it is a topic of interest today where audiovisual media are used on a daily basis, authors still need to work on the article for it to be ready for publication. Some suggestions to improve the manuscript could be:
Title
The title is correct and descriptive of the work.
Introduction
The introduction is appropriate according to the topic addressed. It is descriptive of anxiety and behavior during children's dental treatment, as well as the main techniques to overcome them.
Data on the prevalence of treatments that need to be interrupted or postponed due to children's behavior or anxiety should be entered.
After reviewing the related literature (90manuscript, see references) no percentages have been found for interrupted or postponed treatments due to anxiety or dental fear.
Dental fear/anxiety prevalence in children and adolescence rates were 10.0%,and 12.2%, for the DAS and MDAS scores, respectively. In the studies that used MCDAS Dental fear/prevalence, rates varied from 13.3% to 29.3% in children and adolescents. The complete data synthesis of either Dental Fear prevalence The results were highly heterogeneous, most probably due to the study design, the sampling methods, the setting and the application of the questionnaire, as well as cultural attitudes and socio-economical variations. Up to every fourth adult is reporting dental fears, whereas the point prevalence of clinically relevant dental phobia is estimated to be about 4% [F. Oosterink, A. De Jongh, J. Hoogstraten, Prevalence of dental fear and phobia relative to other fear and phobia subtypes, Eur. J. Oral Sci. 117 (2009) 135–143.]. The prevalence of dental anxiety was 13.3% (95% CI = 11.4, 15.6). Sukumaran I, Taylor S, Thomson WM. The prevalence and impact of dental anxiety among adult New Zealanders. Int Dent J. 2020 Sep 14. doi: 10.1111/idj.12613. Epub ahead of print. PMID: 32929752.
The prevalence rates published in the meta-analysis by Lopez Valverde N in 2021: "Dental fear and anxiety affect approximately 15-20% of children" have been added [Ougradar, A.; Ahmed, B. Patients’ perceptions of the benefits of virtual reality during dental extractions. Br. Dent. J. 2019, 227, 813–816., Hill, K.B.; Chadwick, B.; Freeman, R.; O’Sullivan, I.; Murray, J.J. Adult Dental Health Survey 2009:Relationships between dental attendance patterns, oral health behaviour and the current barriers to dental care. Br. Dent. J. 2013, 214, 25–32. Seligman, L.D.; Hovey, J.D.; Chacon, K.; Ollendick, T.H. Dental anxiety: An understudied problem in youth. Clin. Psychol. Rev. 2017, 55, 25–40.]
The authors note that there are “a handful of studies have used a virtual reality (VR) headset to distract children during dental treatment” [11-13,16]. However, other references included by the authors also use virtual reality for distraction at the dentist. Likewise, I recommend reading the following literature among others:
- Nunna M, Dasaraju RK, Kamatham R, Mallineni SK, Nuvvula S. Comparative evaluation of virtual reality distraction and counter-stimulation on dental anxiety and pain perception in children. J Dent Anesth Pain Med. 2019;19:277-288.
- Niharika P, Reddy N V, Srujana P, Srikanth K, Daneswari V, Geetha K S. Effects of distraction using virtual reality technology on pain perception and anxiety levels in children during pulp therapy of primary molars. J Indian Soc Pedod Prev Dent 2018;36:364-9
- Vabitha Shetty, Lekshmi R Suresh, Amitha M Hegde; Effect of Virtual Reality Distraction on Pain and Anxiety During Dental Treatment in 5 to 8 Year Old Children. J Clin Pediatr Dent 1 2019; 43: 97–102
- Al-Khotani A, Bello LA, Christidis N. Effects of audiovisual distraction on children's behaviour during dental treatment: a randomized controlled clinical trial. Acta Odontol Scand. 2016;74(6):494-501
New bibliographic references have been consulted and have been conveniently added to the manuscript. Thanks for your contribution.
Material and Methods
The methodology used in the clinical trial must be detailed in greater depth, especially as regards dental treatment. The need for preventive, restorative, pulp treatment, extraction or any other treatment is not specified, even knowing the existence of a wide variability between the different treatments for children's anxiety or behavior. Also, an assessment of pain in the different procedures would be desirable.
The Material section has been improved.
On the other hand, an assessment of the patient's previous experiences in dental treatment would be interesting.
The pain has not been evaluated. This subjective experience has not been registered. DMFT index and DFT index are shown in table 1.
Results
The results of the study, together with the statistics analyzed, are presented acceptably.
The results of the DFT and DMFT index are not presented, keys for evaluating the existence of previous treatments.
DMFT index and DFT index have been added in table 1 and in the linear regression model.
Line 148 must be deleted.
Ok, this line has been deleted
Discussion
The discussion shows an interesting analysis of the results of the study in comparison with other studies present in the literature.
A comment with other common techniques to reduce anxiety and improve behavior such as counter-stimulation would be necessary.
New techniques to decrease dental anxiety have been reviewed and added in the Discussion section.
Finally, tables should be modified and references rewritten according to journal guidelines.
OK, Done, Thank you.
Round 2
Reviewer 1 Report
Dear Authors,
Thank you for the opportunity to re-review the paper entitled “Behavior and anxiety management of pediatric dental patients through virtual reality: a randomized clinical trial”. The manuscript's improvement in quality is evident. Congratulations on your engagement. However, the authors still seem to have overlooked several important methodological issues
Major:
- I did not receive information on the registration of the clinical trial. I guess that the authors did not perform this. According to the journal guidelines, every clinical trial should be registered:
“MDPI follows the International Committee of Medical Journal Editors (ICMJE) guidelines which require and recommend registration of clinical trials in a public trials registry at or before the time of first patient enrollment as a condition of consideration for publication… Suitable databases include clinicaltrials.gov, the EU Clinical Trials Register and those listed by the World Health Organisation International Clinical Trials Registry Platform.”
- The authors do not seem to have understood that the randomization procedure should be described in the manuscript and not in a response to reviewers. The quality of scientific research is sometimes judged.... e.g. when someone does a systematic review, I think it's useful to point out the information though...
- I do not find information on sample size in the manuscript. If the authors claim that the included sample is sufficient, please provide statistical calculations to confirm this, based on effect size or minimal clinically important difference (MCID)
Minor:
- Please stick to the established group names: in the text, one can find "experimental group" "test group" or “VR group”.
Author Response
Major:
- I did not receive information on the registration of the clinical trial. I guess that the authors did not perform this. According to the journal guidelines, every clinical trial should be registered:
“MDPI follows the International Committee of Medical Journal Editors (ICMJE) guidelines which require and recommend registration of clinical trials in a public trials registry at or before the time of first patient enrollment as a condition of consideration for publication… Suitable databases include clinicaltrials.gov, the EU Clinical Trials Register and those listed by the World Health Organisation International Clinical Trials Registry Platform.”
R1. We are grateful for this recommendation. To proceed with the registration of the clinical trial, we have sent the Protocol Registration and Results System (PRS) Administrator Contact Request Form (University of Salamanca). ClinicalTrials.gov will create a PRS account within 2 business days of receiving the application. In order not to delay the review process, we send the rest of the reviewers' suggestions answered. We are waiting for a response from the administrator to be able to register the clinical trial soon. In Spain there is no obligation to register the clinical trial beyond the own institutional board, and that is why it has not been carried out previously.
Q2. The authors do not seem to have understood that the randomization procedure should be described in the manuscript and not in a response to reviewers. The quality of scientific research is sometimes judged.... e.g. when someone does a systematic review, I think it's useful to point out the information though...
R2. This phrase has been added in M and M: To effectively randomize the intervention we used sequentially numbered, sealed opaque envelopes, with aleatory numbers that were selected by participant at the first appointment and latter allocated to test (pair numbers) or control (odd numbers) groups accordingly.
- I do not find information on sample size in the manuscript. If the authors claim that the included sample is sufficient, please provide statistical calculations to confirm this, based on effect size or minimal clinically important difference (MCID)
R3. The final sample size was determined using the data distribution and means of the variable dental anxiety of the first 12 participants to calculate that it was needed 38 participants with and power of 0.80% and a alpha error of 0.05% for detecting a true difference in means between the test and the reference group of 0.6.
Minor:
- Please stick to the established group names: in the text, one can find "experimental group" "test group" or “VR group”.
R4. Ok, it has been done. VR group has been fixed.
Reviewer 2 Report
The authors have greatly modified the document, improving the quality of the work. However, although the main questions have been answered to the reviewer, there are serious deficiencies that still need to be considered.
We found a change of authors in a work previously prepared without justification, where it is even indicated that a new author after the review has participated in writing — original draft preparation.
It is not understood that some considerations answered to the reviewers are not included in the manuscript.
Modifications or paragraphs have been introduced in the text, which sometimes do not keep continuity with the manuscript. Also, the new tables do not follow the journal's guidelines.
Finally, although perhaps the most outstanding: after the review there have been changes in the material and methods used: In the original document there is as an inclusion criterion "needing a minimum of 3 appointments to undergo dental treatment (either preventive, conservative, or palliative)" where now we find "needed a minimum of three appointments to undergo conservative dental treatment (fillings)". How is it possible to modify the inclusion criteria after revision without modifying the results?
Author Response
Q1. We found a change of authors in a work previously prepared without justification, where it is even indicated that a new author after the review has participated in writing — original draft preparation.
R1. Cristina Gómez Polo has been added, now she is the first author for its special relevance in the task of revising (major revision) the manuscript (R1 and R2), and making substantial contribution in the interpretation of the findings. All authors of the article agree on the number and order of the authors in the manuscript. Thus, the inclusion criteria as an author are met, according to the rules of the journal.
AUTHOR INSTRUCCIONS OF JCM: Author Contributions: Each author is expected to have made substantial contributions to the conception or design of the work; or the acquisition, analysis, or interpretation of data; or the creation of new software used in the work; or have drafted the work or substantively revised it; AND has approved the submitted version (and version substantially edited by journal staff that involves the author’s contribution to the study); AND agrees to be personally accountable for the author’s own contributions and for ensuring that questions related to the accuracy or integrity of any part of the work, even ones in which the author was not personally involved, are appropriately investigated, resolved, and documented in the literature.
Q2. It is not understood that some considerations answered to the reviewers are not included in the manuscript.
R2. We have now included previous comments in the paper
Q3. Modifications or paragraphs have been introduced in the text, which sometimes do not keep continuity with the manuscript.
R3. From our point of view, the next paragraphs do not maintain expository continuity and has been deleted. Please let us know if it is correct.
Q4. Also, the new tables do not follow the journal's guidelines.
R4. To our knowledge and using the JCM template the tables follow the journal style
Q5. Finally, although perhaps the most outstanding: after the review there have been changes in the material and methods used: In the original document there is as an inclusion criterion "needing a minimum of 3 appointments to undergo dental treatment (either preventive, conservative, or palliative)" where now we find "needed a minimum of three appointments to undergo conservative dental treatment (fillings)". How is it possible to modify the inclusion criteria after revision without modifying the results?
R5. The inclusion criteria of the participants have not been modified. At first, the inclusion criterion that appeared in the manuscript was “needing a minimun of 3 appointments to undergo dental treatment (either preventive, conservative or palliative)”. After reviewing all the treatments performed on the participants we realized that all the treatments performed were fillings. None of the treated children needed palliative or preventive treatments, only fillings of distinct deepness were performed. In order to be more rigorous in explaining the treatments performed and not confuse the reader, it has been decided to eliminate "preventive and palliative treatments". Please, if you need any further explanation, let us know, we will be delighted.
Thank you for your good work as a reviewer. We remain at your disposal to improve the text in the best way you consider appropriate.